# A New Side-Looking Scheme for Speed Estimation and Detection of Tangential Slow-Moving Targets

**DOI:** 10.3390/s22124535

**Published:** 2022-06-16

**Authors:** Ziyi Qi, Xiaohong Huang, Lanpu He

**Affiliations:** 1College of artificial intelligence, North China University of Science and Technology, Tangshan 063210, China; qzy971031@gmail.com (Z.Q.); iehlphlp@163.com (L.H.); 2Hebei Key Laboratory of Industrial Intelligence Perception, Tangshan 063210, China

**Keywords:** frequency-modulated continuous wave (FMCW), intelligent transportation system (ITS), measurement, target detection, channel probing

## Abstract

A single-beam radar system cannot adopt a side-looking installation scheme, which is completely perpendicular to the moving direction of the target in an intelligent transportation system (ITS), because of its own limitations. In this paper, a side-looking radar velocity measurement system that utilizes a new signal processing method and multi-channel radar scheme is proposed. Constant false alarm rate (CFAR) and generalized likelihood ratio test (GLRT) detectors are used to detect the data processing results in different stages in order to reduce the false alarm rate of targets. At the same time, a deconvolution-based clutter map algorithm is proposed to solve the problem of clutter interference in the test environment, and its theoretical performance is verified by simulation. Finally, 77 G commercial radar is used to test the system, and the results show that this algorithm can effectively detect and accurately estimate the speed of tangential low-speed targets under clutter interference.

## 1. Introduction

The basic function of modern intelligent transportation systems is to detect and measure the speed of the target in the observation area. In addition to traditional solutions such as piezoelectric sensors, ultrasonic and infrared sensors [1], and video detectors [2,3,4,5], radar sensors with relatively mature technology in military applications have previously also been applied in intelligent transportation systems (ITSs), river information services (RISs) [6,7,8], and other fields. Compared with traditional methods, radar sensors have the advantages of small size, light weight, low cost, and all-weather operation. Frequency-modulated continuous wave (FMCW) radar, which is widely used in traffic data collection, can measure speed by transmitting continuous waves (CWs) and comparing the beat frequency [9] or displacement difference [10] between the transmitted and received signals. As is known, the radial velocity of a moving object usually causes the frequency of the received signal to deviate from the carrier frequency, which is called the Doppler frequency [11,12]. The measurement of target velocity by this method is based on the Doppler frequency of the target [13]. However, the Doppler frequency is insensitive to targets with only a tangential velocity or weak radial velocity [14], the reason being that the velocity of a moving object is a vector, and the position of the target relative to the observation radar will affect the radial and transverse components of its velocity. Therefore, when the radial velocity of the target is weak or the moving direction is perpendicular to the line of sight (LOS) of the radar, the radar cannot effectively estimate the target velocity by the Doppler frequency. In order to avoid the above problems, the radar used in traffic systems is often installed at the top of gates or bridges, and the line of sight of radar is parallel to the road. However, the disadvantage of this is that the radar monitoring range is limited [15] and the LOS of the radar is easily blocked by obstacles [16]. Therefore, the side-looking scheme perpendicular to the monitored lane is the preferred installation scheme. Traditional side-looking radar uses two schemes to estimate the target velocity. The first scheme estimates the velocity by measuring the time when the target stays in the radar illumination area. To realize this scheme, it is necessary to use special radar equipment to transmit a wide beam to improve the measurement effect of the weak Doppler frequency [17,18]. However, this scheme will not only sacrifice ranging accuracy, but also is not suitable for measuring long-range targets and is more likely to cause ghost targets by multipath effects [19], so it is not conducive to practical application. The second scheme adopts dual-radar detectors. The time between two radar beams with large horizontal spacing can estimate the target velocity by correlation or convolution, but this scheme also has great limitations. The radar equipment needs to be installed with high precision to better match the received signals between the two antennas. Moreover, complex traffic scenes will seriously affect the measurement. Therefore, the recent research uses a narrow-beam single-radar detector and considers introducing a small tilt angle, then carries out range-Doppler processing to measure the target velocity [20]. However, there are some problems, such as velocity ambiguity, the increased time of distant targets in the radar beam, and the decrease of velocity measurement accuracy. Other studies considered velocity measurement based on the Doppler history [21] of moving targets and the correlation between azimuth and Doppler frequency shift [22], but this involves complex hardware and signal processing, so the cost is relatively high. For example, the system used in [22] has one transmitting antenna and two receiving antennas, which are separated by 7.5 cm. Similar methods in other fields also need a wide spacing between the receiving antennas [23,24,25,26], which is difficult to achieve for commercial millimeter wave radars. In addition, this scheme takes a long time to obtain the Doppler frequency shift and the azimuth change of slow-moving targets, which increases the difficulty of processing and application, so further consideration needs to be made to improve the computational efficiency [27]. To address the above issues, this paper uses a novel radar system to detect the target moving perpendicular to the LOS direction and focuses on the velocity measurement of the slow target. Compared with faster-moving vehicles, slow targets are more difficult to detect by the radar system, because the Doppler frequencies of slow targets are usually close to the clutter spectrum, which means the moving speed cannot be accurately estimated [28,29]. The main technical contributions of this paper are summarized as follows:
A new side-looking radar scheme is proposed, which uses the change in frequency of the wave-path difference between the receiving channels to replace the Doppler frequency to estimate the velocity of the tangential moving target; the scheme can ensure the accuracy of the velocity measurement and reduce the processing complexity.A deconvolution-based clutter map algorithm is proposed to solve the problem of slow and weak targets being susceptible to clutter, and this algorithm can pre-learn the clutter environment to reduce the computation and running time in actual measurements.The theoretical performance and performance curve of the detector in the system are given, and the performance of the detector before and after clutter processing is simulated to verify the effectiveness of the proposed clutter processing algorithm.The scheme of this paper has no special requirements for radar equipment. Theoretically, the scheme can be deployed on any millimeter-wave radar equipment with multiple channels. In addition, this paper solves the problem that the phase difference is not obvious due to the small spacing of the receiving channels and verifies it on commercial millimeter-wave radar equipment. 

The remainder of this article is organized as follows. The theoretical derivation and simulation of the proposed velocity measurement method are carried out in Section 2. The theoretical performance and simulation results of the detector are provided in Section 3. In Section 4, the treatment scheme for the clutter interference of the stationary target is introduced. In Section 5, the actual test results are given. Finally, the research is summarized in Section 6.

## 2. Proposed Method

Figure 1 shows the installation position of the radar system and the motion mode of the target. The target passes through the radar detection area at a certain speed along the direction perpendicular to the radar LOS, and the radar uses two receiving channels with a distance of *D* to receive the echo signal. Due to the range change of the target relative to the side-looking radar system usually not being obvious, the triangular wave will reduce the pulse repetition frequency and affect the detection of the Doppler frequency, so the signal transmission waveform adopts a sawtooth wave.

In order to facilitate the theoretical derivation, the target is simplified as a point model. The modulated LFM signal transmitted by the radar can be expressed as: (1)$$s\left(\hat{t},t_{m}\right)=\exp\left(j\pi \gamma {\hat{t}}^{2}\right)\exp\left(j2\pi f_{c}t\right)$$
where −T/2≤t≤T/2, *T* is the pulse repetition interval (PRI), and γ is the frequency modulation slope.

The signal propagates after being reflected by the target and is received by the receiving antenna. The echo delay is related to the target distance. In general, when the pulse repetition period emitted by the radar is much longer than the pulse duration time and the pulse repetition frequency is much larger than the Doppler frequency of the target, the target can be approximately considered not to move in the pulse duration and the distance between the target and the radar does not change. Then, the echo signal, which is received by the radar, is down-converted, and the result is as follows: (2)$$x_{m}\left(t\right)=z\exp\left(j\pi \gamma {\left(\hat{t}-\tau_{m}\right)}^{2}\right)\times \exp\left(-j2\pi f_{c}\tau_{m}\right)+\tilde{u}\left(t\right)$$
where $\tau_{m}=2\left(r\left(t_{m}\right)/c\right)$; it is the echo delay of transmitting pulses that is transmitted at different times. $r\left(t_{m}\right)$ is the distance between the radar and target when the pulses reach the target. $\tilde{u}\left(t\right)$ is the Gaussian white noise. $f_{c}$ is the carrier frequency. *c* is the speed of light. *z* is the target reflection coefficient.

If there are multiple targets at the same time, the echo signal received by the radar will be the superposition of echo signals of different targets, and the processed echo signal can be expressed as: (3)$$x_{m}\left(t\right)=\sum_{i=1}^{I}z_{i}\exp\left(j\pi \gamma {\left(\hat{t}-\tau_{i,m}\right)}^{2}\right)\times \exp\left(-j2\pi f_{c}\tau_{i,m}\right)+\tilde{u}\left(t\right)$$

To sample the received signal $x_{m}\left(t\right)$, the sampling frequency $f_{s}$ should conform to the sampling theorem. The echo signal of the *m* pulse emitted by the sampled radar can be expressed as: (4)$${\tilde{x}}_{m}\left(n\right)=\sum_{i=1}^{I}z_{i}\tilde{s}\left(n-f_{s}\left(\tau_{i,m}-T/2\right)\right)\times \exp\left(-j2\pi f_{c}\tau_{i,m}\right)+{\tilde{u}}_{m}\left(nT_{s}-T/2\right)$$
where $\tilde{s}\left(n\right)=s\left(nT_{s}-T/2\right)$ is the sampling signal. $N_{w}=f_{s}T$ is the number of sampling points. $T_{s}=1/f_{s}$ is the sampling interval. *n* is the sampling sequence number, $0\leq n\leq N_{w}$.

Perform $N_{F}$ point fast Fourier transform (FFT) on ${\tilde{x}}_{m}\left(n\right)$, and if the number of samples is not enough, fill zeros after ${\tilde{x}}_{m}\left(n\right)$; its spectrum can be obtained as follows:(5)$$\begin{array}{cc} X_{m}\left(n\right) & =\sum_{n=0}^{N_{f}-1}{\tilde{x}}_{m}\left(n\right)\exp\left(-j2\pi nk/N_{f}\right)\\ \; & =\sum_{i=1}^{I}z_{i}\exp\left(-j2\pi f_{c}\tau_{i,m}\right)S\left(k\right)\times \exp\left(-j2\pi kf_{s}\left(\tau_{i,m}-T/2\right)/N_{f}\right)\;+U_{m}\left(k\right) \end{array}$$

$X_{m}\left(k\right)$ and S(k) are conjugated and then multiplied. Because there is a time difference between the signal starting time and the sampling starting time, the result after signal-matched filtering can be obtained after the corresponding compensation as:(6)$$\begin{array}{cc} {\bar{X}}_{m}\left(k\right) & =X_{m}\left(k\right)S^{*}\left(k\right)\times \exp\left(-j2\pi k\Delta fT/2\right)\\ \; & =\sum_{i=1}^{I}{\bar{z}}_{i}\exp\left(-j2\pi k\Delta f\tau_{i,m}\right)\times \exp\left(-j2\pi f_{c}\tau_{i,m}\right)+{\bar{U}}_{m}\left(k\right) \end{array}$$

The radar system consists of two receiving antennas spaced *D*, as shown in Figure 2. Assuming that the aperture of the array element is far less than the distance between the signal source and the center of the array, the spherical wave emitted by the point emitter can be regarded as a plane wave. At this time, the difference of the incoming wave direction between the two antennas is very small, which can be approximately equal to θ. However, due to the different positions of the two antennas, there is a difference in the wave path between the receiving antennas, which leads to a slight difference in the phase of the received signal. Therefore, the angle and moving speed of the target can be estimated.

As can be seen from Equation (Equation 6), the output of the received signal of RX1 after matched filtering can be expressed as:(7)$$\begin{array}{c} X_{m,k}^{1}=z_{1}\exp\left(-j2\pi k\Delta f\left(\tau_{m}+\tau_{m}^{d}\right)\right)\times \exp\left(-j2\pi f_{C}\left(\tau_{m}+\tau_{m}^{d}\right)\right)+{\hat{U}}_{m,k}^{1} \end{array}$$
where $\tau_{m}^{d}$ is the delay difference caused by the difference of the wave path. Similarly, the output of RX2 after matched filtering can be expressed as:(8)$$\begin{array}{c} X_{m,k}^{2}=z_{2}\exp\left(-j2\pi k\Delta f\tau_{m}\right)\times \exp\left(-j2\pi f_{C}\tau_{m}\right)+{\hat{U}}_{m,k}^{2} \end{array}$$

The signals of the two antennas are conjugated and cross-correlated, and the influence of different frequencies is ignored. The simplified results are as follows:(9)$$\begin{array}{c} X_{m,k}^{1}{\hat{X}}_{m,k}^{2}=z_{2}z_{2}\exp\left(-j2\pi f_{r}\tau_{m}^{d}\right)+{\hat{U}}_{m,k}^{\prime } \end{array}$$

From Equation (Equation 9), it can be seen that the phase change of the two receiving antennas after cross-correlation is a sine wave, which was verified by simulation experiments with the radar parameters in Table 1.

The simulation experiment simulates the target moving at a certain speed along the LOS direction perpendicular to the radar. FFT is performed on the dual-channel received signal after cross-correlation processing, and the result is shown in Figure 3. The simulation results verify that the characteristic signal of the tangential moving target echo signal after processing is a sine wave with a certain frequency, which can realize the velocity estimation of the target and provide a basis for subsequent detection.

The estimation method of the moving speed of the tangentially moving target is given by the following formula: (10)$$f\approx \frac{VD}{L\lambda }$$
where *f* represents the signal frequency after cross-correlation processing, *D* is the distance between different receiving channels of the radar system, *L* is the projection distance between the tangential moving target and receiving array element, and λ is the wavelength of the carrier signal.

## 3. Target Detection

### 3.1. Detect Targets in Different Distance Units

The target detection is divided into two stages. Firstly, the range dimension of the echo signal after processing is detected to remove the range unit without an obvious target and obviously only clutter. The detection problem of the distance dimension can be described as follows: assuming that the transmitted signal x(t) is a chirp signal, the received signal $x_{r}\left(t\right)$ can be expressed as: (11)$$x_{r}\left(t\right)=Ax\left(t-t_{0}\right)\exp\left(j2\pi f_{dop}t\right)+n\left(t\right)$$
where *A* represents the complex amplitude of the signal. After the cross-correlation of the received signal and transmitted signal, the signal with low-pass filtering can be expressed as: (12)$$y\left(t\right)=A\exp\left(j2\pi F\left(t_{0}\right)+f_{dop}\right)t+n_{2}\left(t\right)$$
where $n_{2}\left(t\right)$ is the noise part after the cross-correlation between $x_{r}\left(t\right)$ and x(t). Due to the characteristics of complex Gaussian noise itself, $n_{2}\left(t\right)$ is still an independent and identically distributed complex Gaussian noise. $F(t_{0})$ denotes the signal frequency offset due to time delay. After the fast Fourier transform (FFT), Doppler compensation, and the clutter processing algorithm mentioned above for the mixed signal, there will be a peak value at the corresponding position of the target. The detection problem here considers using the traditional constant false alarm rate (CFAR) detector for target detection, and the decision criteria of the detector are as follows: (13)$$\left\{\begin{array}{c} H_{1}:x_{i}\geq \alpha {{\hat{\sigma }}^{2}}_{w}\\ H_{0}:x_{i}\leq \alpha {{\hat{\sigma }}^{2}}_{w} \end{array}\right.$$
where $H_{1}$ represents the presence of a target in the detection unit; $H_{2}$ denotes that there is no target in the detection unit; α represents the threshold product factor; ${{\hat{\sigma }}^{2}}_{w}$ is the estimated value of the power level in the clutter background, and the product of them is the threshold decision threshold.

### 3.2. Detect the Characteristic Signal of Tangential Moving Target

In the second stage, the characteristic signal of a tangential moving target, which is obtained after cross-correlation processing of two-channel received signals, is detected. The previous section proved that the processed signal is a sine wave with a fixed frequency, so the detection problem here can be regarded as a sine signal detection problem in a noisy environment, which can be expressed as: (14)$$z\left(t\right)=\left\{\begin{array}{c} A\exp\left(j2\pi f_{r}\tau_{m}^{d}\right)+n_{2}\left(t\right),H_{1}\\ n_{2}\left(t\right),H_{0} \end{array}\right.$$

Assuming *n* is the normalized noise power, the probability density function of the correlated received signal can be expressed as:
In the case of a null hypothesis:


(15)
$$f\left(z\left(t\right)\left|H_{0}\right.\right)=\frac{1}{\sqrt{2\pi }}\exp\left(-\frac{{\left(z(t)\right)}^{2}}{2}\right)$$


In the case of an alternative hypothesis:


(16)
$$f\left(z\left(t\right)\left|H_{1},A,f_{r}\right.\right)=\frac{1}{\sqrt{2\pi }}\exp\left(-\frac{{\left(z\left(t\right)-A\exp\left(j2\pi f_{c}t\right)\right)}^{2}}{2}\right)$$


The generalized likelihood ratio detector at this time can be expressed as (the amplitude, phase, and frequency of signal are unknown): (17)$$c=\arg\max_{f_{r}}{{\left|\sum z(t)\exp(-j2\pi f_{r}t)\right|}^{2}}_{<}^{>}\gamma_{2}$$

The false alarm probability and detection probability of the detector can be expressed as: (18)$$P_{f}=1-{\left(1-\int_{\gamma }^{\infty }f(c\left|H_{0}\right.)dc\right)}^{N}$$
(19)$$P_{d}=P_{d_{1}}\left(1-P_{f}\right)+P_{f}\left(1-P_{d_{1}}\right)$$
where $P_{d1}=\int_{\gamma }^{\infty }\int f(c\left|H_{1},A\right.)f\left(A\right)\mathrm{dAd}c$.

The simulation experiment still uses the radar parameters in Table 1 and simulates the detection of the target characteristic signal that was obtained after cross-correlation calculation of the two-channel echoes under a background of complex Gaussian noise. The theoretical detection performance and simulation detection performance of tangentially moving targets in a noisy environment under different signal-to-noise ratios can be obtained as shown in Figure 4.

Simulation experiments show that the detector can effectively detect the characteristic signal of the target in a noisy environment, but there are various interferences such as clutter in the actual test environment, which easily affect the performance of the detector and need to be considered.

## 4. The Method of Dealing with Clutter

Since the radial Doppler frequency of the tangential slow-moving target is weak and its frequency spectrum is close to the clutter of a stationary target, it is easily interfered with by clutter in the environment, which leads to the inability to detect and measure the velocity of this kind of target effectively. Although some obvious clutter can be removed by target detection and screened in the distance dimension at the beginning, signal detection is still vulnerable to clutter interference under the conditions of slow target movement and a low signal-to-noise ratio (SNR), as shown in Figure 5 and Figure 6. Therefore, it is necessary to deal with clutter to improve the signal-to-noise ratio and ensure the reliability of the detection effect.

Clutter mapping is an effective method to eliminate clutter interference in the radar detection area when the clutter changes slowly in the time domain and dramatically in the space domain. Its main idea is to iterate and record the clutter background in the same resolution unit through multiple scanning periods and remove the clutter background after iteration in the corresponding unit in the subsequent signal processing. The iterative mode can be expressed as [30]: (20)$${\hat{p}}_{n}\left(k\right)=\left(1-\omega \right){\hat{p}}_{n-1}\left(k\right)+\omega q_{n}\left(k\right)$$
where ${\hat{p}}_{n}\left(k\right)$ represents the estimated clutter value at the *n* resolution unit during the *k* scan; $q_{n}\left(k\right)$ denotes the clutter estimation at the *n* resolution unit at the *k* scan; ω denotes the iterative forgetting factor.

The clutter map can effectively suppress the clutter that changes slowly with time in the detection area. However, it is difficult to remove the remaining clutter and noise in the transmission channel at the same time only by iteratively updating the recorded amplitude. Since the equivalent channel response of static clutter can be considered as the superposition of multiple different impulse responses, this section proposes a clutter map algorithm based on deconvolution to estimate the equivalent channel response and cancel the clutter, so as to solve the problems existing in the traditional clutter map algorithm. The pseudo code of the proposed algorithm is summarized in Algorithm 1.
**Algorithm 1** Clutter map algorithm based on deconvolution**Input:** transmit signal $x_{i}\left(t\right)$; received signal $y_{i}\left(t\right)$; iterative forgetting factor ω
**Output:** signal after clutter removal $z_{i}\left(t\right)$
1:initial $S_{d}^{\prime }\left(t\right)$ =null;2:**for**i=1;i≤n**do**3:    **if** i=1||i reaches a certain number of iterations **then**4:          equivalent channel response $h^{\prime }\left(t\right)=F^{-1}\{\frac{{R^{\prime }}_{xy}\left(f\right)}{{R^{\prime }}_{xx}\left(f\right)}\}$5:          background clutter $S_{d}\left(t\right)=x\left(t\right)*h^{\prime }\left(t\right)$6:          $z_{i}\left(t\right)=y\left(t\right)-S_{d}\left(t\right)$7:          $S_{d}^{\prime }\left(t\right)=S_{d}\left(t\right)$8:    **else**
$z_{i}\left(t\right)=y\left(t\right)-\left[\omega \times S_{d}\left(t\right)+\left(1-\omega \right)\times S_{d}^{\prime }\left(t\right)\right]$9:**return**$z_{i}\left(t\right)$


It is worth noting that the clutter background in the experimental environment usually does not change dramatically with time. Therefore, considering the algorithm complexity and computation, the deconvolution of the output from the clutter map is only carried out after a certain number of iterations or in the initial data. The specific steps of deconvolution processing in the algorithm are shown in Figure 7. Assuming that the transmitted signal is x(t), the received signal of the channel can be expressed as: (21)$$y\left(t\right)=s_{r}\left(t\right)+s_{d}\left(t\right)+\sqrt{P_{n}}\times n\left(t\right)$$
where $s_{r}\left(t\right)$ is the received target reflection signal, $s_{d}\left(t\right)$ is the clutter interference signal, n(t) is the noise, and $P_{n}$ is the noise power. $s_{r}\left(t\right)$ and $s_{d}\left(t\right)$ can be expressed as:
(22)$$s_{r}\left(t\right)=\sqrt{P_{r}^{r}}\left(h\left(t-\tau_{d}\right)*x\left(t-\tau_{d}\right)\right)e^{j2\pi \left(f_{d}+f_{a}\right)t+j\phi^{r}}$$
(23)$$s_{d}\left(t\right)=\sqrt{P_{r}^{d}}\left(h\left(t\right)*x\left(t\right)\right)e^{j2\pi f_{a}t+j\phi^{d}}$$
where $P_{r}^{r}$ and $P_{r}^{d}$ are, respectively, the power of the corresponding received signal component, $\phi^{r}$ and $\phi^{d}$ are the random phase shift caused in the transmission process, $f_{a}$ is the intermediate frequency after the received signal is mixed with the local reference signal, $\tau_{d}$ is the reflected signal time delay caused by the target position, $f_{d}$ is the Doppler frequency offset caused by the target motion, and h(t) is the channel transmission response from the target to the receiving service.

It can be seen from Equation (Equation 23) that there is a difference between the radar transmission signal x(t) and the clutter interference signal $s_{d}\left(t\right)$, where $e^{j2\pi f_{a}t}$ includes the frequency domain difference between the received signal and the reference signal. Because the frequency offset of the clutter interference signal is usually easy to estimate, the frequency domain difference can be used to effectively eliminate $s_{d}\left(t\right)$ in the received signal. Then, the frequency offset in Equation (Equation 23) can be rewritten as: (24)$$s_{d}^{\prime }\left(t\right)=\left(\sqrt{P_{r}^{d}}e^{j\phi^{d}}h\left(t\right)\right)*x\left(t\right)=h^{\prime }\left(t\right)x\left(t\right)$$
where $h^{\prime }\left(t\right)$ is the equivalent channel response during signal transmission. It can be seen from Equation (Equation 24) that in order to accurately reconstruct $s_{d}^{\prime }\left(t\right)$ to cancel clutter interference signals, it is necessary to estimate the transmission response $h^{\prime }\left(t\right)$ in the time domain. The deconvolution method is considered to recover the transmission response $h^{\prime }\left(t\right)$.

Considering the noise in the received signal, the cross-correlation calculation pretreatment is carried out for the input signal of each channel and the received signal y(t) by using the reference signal x(t), and the results can be respectively expressed as: (25)$$R_{xx}\left(t\right)=\int x(\tau )x(\tau -t)d\tau$$
(26)$$\begin{array}{cc} R_{yx}\left(t\right) & =\int y(\tau )x(\tau -t)d\tau \\ \; & =h^{\prime }\left(t\right)*R_{xx}\left(t\right)+\sqrt{P_{n}}R_{n^{\prime }x\left(t\right)} \end{array}$$
where $R_{yx}\left(t\right)$, $R_{xx}\left(t\right)$, and $R_{n^{\prime }x}\left(t\right)$ represent the result of the cross-correlation calculation between the reference signal x(t) and the received signal y(t), the transmitted signal x(t) and $n^{\prime }\left(t\right)$ of each channel, respectively. At the same time, because the noise signal $n^{\prime }\left(t\right)$ and the signal x(t) are independent of each other, when the signal length is long enough, the result $R_{n^{\prime }x}\left(t\right)\;=\;0$ after cross-correlation can be considered, and at this time, $R_{n^{\prime }x}\left(t\right)$ can be ignored.

Therefore, it can be seen that the equivalent response $h^{\prime }\left(t\right)$ of the time domain transmission channel can be recovered by deconvolution of the result $R_{yx}\left(t\right)$ after cross-correlation, and its expression can be written as: (27)$${\tilde{h}}^{\prime }\left(t\right)=R_{yx}\left(t\right)*F^{-1}\left\{\frac{1}{F\left\{\left\{R_{xx}\left(t\right)\right\}\right\}}\right\}$$

The deconvolution step in Equation (Equation 27) can be implemented in the frequency domain, and the algorithm implementation flow is shown in Figure 8.

The specific implementation process is as follows: the frequency domain representations $R_{yx}\left(f\right)$, $R_{xx}\left(f\right)$, and $R_{n^{\prime }x}\left(f\right)$ can be obtained from the cross-correlation results $R_{yx}\left(t\right)$; $R_{xx}\left(t\right)$ and $R_{n^{\prime }x}\left(t\right)$ of each signal can be obtained through FFT; the equivalent channel response at this time can be expressed as: (28)$$H^{\prime }\left(f\right)=R_{yx}\left(f\right)/R_{xx}\left(f\right)-\sqrt{P_{n}}R_{n^{\prime }x}\left(f\right)/R_{xx}\left(f\right)$$

After the cross-correlation calculation of the transmitted signal x(t) and the channel received signal y(t), $R_{yx}\left(t\right)$ will only have a peak value at the sampling point near, so it can be considered that the signal strength at this time is greater than the noise strength. Therefore, intercepting and zeroing the cross-correlation results between signals is helpful to further suppress the noise, and the corresponding processing can be expressed as: (29)$$R_{yx}^{\prime }\left(t\right)=\left\{\begin{array}{c} R_{yx}\left(t\right),\tau^{\prime }-k<\tau <\tau^{\prime }+k\\ 0,\mathrm{otherwise} \end{array}\right.$$
(30)$$R_{xx}^{\prime }\left(t\right)=\left\{\begin{array}{c} R_{xx}\left(t\right),-k<\tau <k\\ 0,\mathrm{otherwise} \end{array}\right.$$
where *k* is the truncation boundary of the cross-correlation result, whereby the equivalent response to the signal transmission channel can be expressed as:(31)$$\hat{H}\left(f\right)=\frac{{R^{\prime }}_{yx}\left(f\right)}{{R^{\prime }}_{xx}\left(f\right)}=\frac{R_{yx}\left(f\right)*W\left(f\right)}{R_{xx}\left(f\right)*W\left(f\right)}$$
where W(f) represents the frequency domain representation of the truncated rectangular window. At the same time, due to the corresponding H(f) of channel transmission not varying drastically in the signal frequency band, Equation (Equation 31) can be written as:(32)$$\begin{array}{c} \tilde{H}\left(f\right)=\frac{H\left(f\right)(R_{xx}\left(f\right)*W\left(f\right))}{R_{xx}\left(f\right)*W\left(f\right)}+\frac{R_{N^{\prime }x}\left(f\right)*W\left(f\right)}{R_{xx}\left(f\right)*W\left(f\right)}\\ =H\left(f\right)+\frac{R_{N^{\prime }x}\left(f\right)*W\left(f\right)}{R_{xx}\left(f\right)*W\left(f\right)} \end{array}$$

The second term of the result in Equation (Equation 32) is the noise term. According to the Parseval theorem, the noise power after processing is reduced to k/N-times the original, where *N* is the sampling length of the signal. Therefore, the noise interference is greatly reduced and the channel transmission response $h^{\prime }\left(t\right)$ is estimated. Finally, the signal processed by the deconvolution method can be obtained as follows: (33)$$z\left(t\right)=y\left(t\right)-s_{d}^{\prime }\left(t\right)=y\left(t\right)-h^{\prime }\left(t\right)*x\left(t\right)$$

Through the method above, we can effectively remove the clutter in the detection area and effectively improve the signal-to-noise ratio. The moving target is detected under the same simulation conditions as Figure 5 and Figure 6, and the detection results are shown in Figure 8 and Figure 9. It can be seen from Figure 8 that the target signal that cannot be detected in Figure 5 is very obvious, and the peak value of the signal exceeds the threshold by about 7 dB.

Figure 10 shows the magnitude of the improvement of the detection performance for tangentially moving targets after the signal is cluttered. After the clutter processing, when the signal-to-noise ratio is −4 dB, the target detection performance is improved by nearly −3 dB. The results prove that clutter processing effectively improves the detection performance of the target characteristic signals under the conditions of a low signal-to-noise ratio and strong clutter interference.

## 5. Experimental Results

In the realistic experimental scene, the test equipment adopts IWR1443 millimeter-wave radar, which is a commercial radar system of Texas Instruments, and adopts DCA1000 for real-time data acquisition, as shown in Figure 11. IWR1443 millimeter-wave radar has three transmit channels (one of which is an elevation antenna) and four receive channels with spacing, which can form a standard virtual array with eight receive channels for target azimuth estimation. In order to simulate strong static clutter interference, the experiment sets a static corner reflector in the radar detection area, and then, pedestrians pass through the radar detection area at a constant speed from outside the radar detection area along the position perpendicular to the radar line of sight of about 10 m. The radar parameters used for data acquisition are shown in Table 2.

Compared with radar equipment with a large receiving channel spacing, it is difficult to detect and measure tangential targets, because the spacing between different receiving channels in the radar antenna array is very small, so the phase difference between the different receiving channels is very small. In this case, a more complicated treatment must be carried out.

First, the method we propose needs to integrate the signal for a longer time. In addition, in order to improve the signal-to-noise ratio of the target signal, the method also estimates, reconstructs, and reduce the noise of the signal. The specific methods are as follows, assuming that the received signal antennas 1 and 2 are: (34)$$y_{1}\left(t,\tau \right)=A_{1,1}\left(\tau \right)e^{j2\pi f_{1}\left(\tau \right)t}+\sum_{i=2}^{L}A_{i,1}\left(\tau \right)e^{j2\pi f_{i}\left(\tau \right)t}+n\left(t,\tau \right)$$
(35)$$y_{2}\left(t,\tau \right)=A_{1,2}\left(\tau \right)e^{j2\pi f_{1}\left(\tau \right)t}+\sum_{i=2}^{L}A_{i,2}\left(\tau \right)e^{j2\pi f_{i}\left(\tau \right)t}+n\left(t,\tau \right)$$
where *t* denotes fast time, τ denotes slow time, and *L* denotes the number of targets to be detected. In Equation (Equation 35), the first term is the signal to be detected, and the phase difference between the two antennas is also included in $A_{1,1}$ and $A_{1,2}$.

The method calculates the FFT for each frame of data and preliminarily screens and delimits the target range according to the detection results in the distance dimension. Because different distances will lead to different frequency offsets of reflected signals, we need to obtain the maximum frequency $f_{est}\left(t\right)$ corresponding to the target position. Each frame is calculated to ensure that the target is not lost due to the slight change in the radial distance of the tangential moving target.

Then, the method reconstructs the received signals of RX1 and RX2, which is an effective method to improve the signal-to-noise ratio.
(36)$$y{}_{1}^{\prime }\left(t,\tau \right)=Y_{1}\left(f_{est}\left(\tau \right),\tau \right)e^{j2\pi f_{est}\left(\tau \right)t}$$
(37)$$y{}_{2}^{\prime }\left(t,\tau \right)=Y_{2}\left(f_{est}\left(\tau \right),\tau \right)e^{j2\pi f_{est}\left(\tau \right)t}$$

In addition, the method correlates the signals of the two receiving channels after reconstruction, then removes the direct-current component and performs a Fourier transform on the data of each fast-time sampling point.
(38)$$y_{12}\left(t,\tau \right)=y{}_{1}^{\prime }\left(t,\tau \right)*conj\left(y{}_{2}^{\prime }\left(t,\tau \right)\right)$$
(39)$$y{}_{12}^{\prime }\left(t,\tau \right)=y_{12}\left(t,\tau \right)-\int_{\tau }y_{12}\left(t,\tau \right)d\tau$$
(40)$$Y{}_{12}^{\prime }\left(t,f\right)=F\left(y{}_{12}^{\prime }\left(t,\tau \right)\right)$$

Finally, $y_{12}$ is averaged in the fast-time dimension to further improve the SNR.
(41)$$\theta \left(f\right)=∥\int_{t}Y{}_{12}^{\prime }\left(t,f\right)dt∥$$

In Figure 12, in order to fully show the process of the target moving perpendicular to the LOS of the radar system, we retained the stationary target and the moving target when it is completely perpendicular to the LOS of the radar system. The speed of the target varies with the relative position between the target and the radar and is far less than the true speed of the target. When the moving direction of the target is perpendicular to the LOS direction of the radar, it can be seen that the system cannot estimate its velocity.

Figure 13 shows the processing results of moving targets. The system estimates that the speed of the slow tangential moving targets is about 1.2 m/s, which is close to the real speed of the target (about 1.5 m/s, obtained by the distance of the target moving in a certain time).

Figure 14 is the processing result of the system for the target with strong clutter interference. It can be seen that the system can accurately identify the strong clutter interference target at 15.6 m and completely remove this target, which has a good clutter processing effect and proves the performance and reliability of the system.

## 6. Conclusions

In this paper, a new processing scheme was proposed to measure the velocity and detect the tangential moving target relative to the radar system’s line of sight direction. This paper focused on the speed measurement and detection of relatively slow-moving weak targets such as pedestrians, because such targets are more susceptible to clutter interference. This paper also proposed a clutter map algorithm based on deconvolution to solve the problem of clutter interference. Simulation results showed that after clutter processing, the detection performance of the detector was improved by about −3 dB when the signal-to-noise ratio was −4 dB. In addition, another advantage of this method is that it can predict and record the clutter around the radar detection when conditions permit, so as to reduce the processing time of the algorithm and improve the clutter processing effect. Finally, this paper overcame the problem that the phase difference between different channels of equipment with small receiving antenna spacing does not change obviously and used the commercial radar system IWR1443 to achieve an accurate velocity estimation of the target. The velocity error of the tangential slow-moving target estimated by the system is less than 0.3 m/s, which proves the effectiveness and reliability of the whole system.

## Figures and Tables

**Figure 1 sensors-22-04535-f001:**
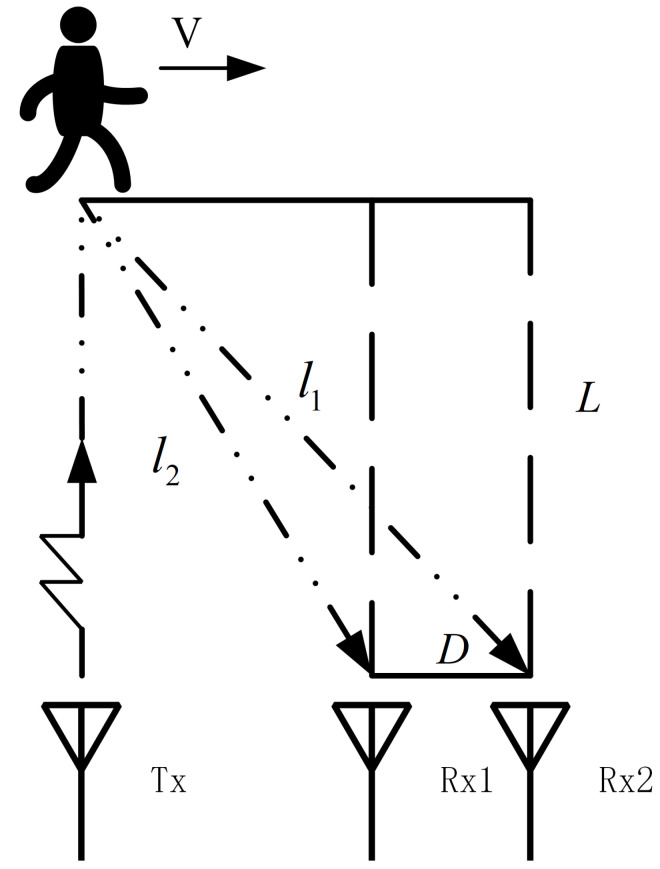
Measurement scheme.

**Figure 2 sensors-22-04535-f002:**
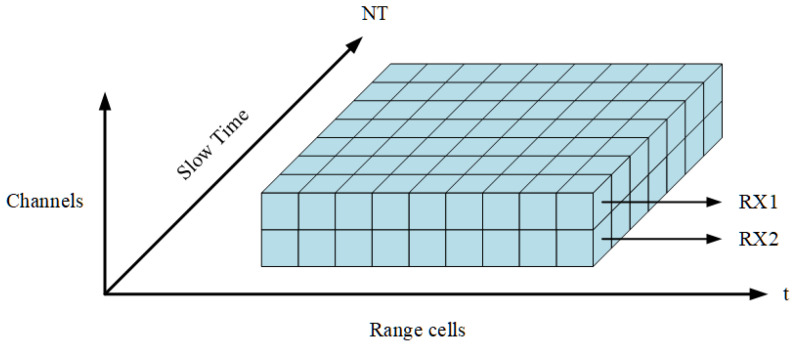
Radar processor datacube.

**Figure 3 sensors-22-04535-f003:**
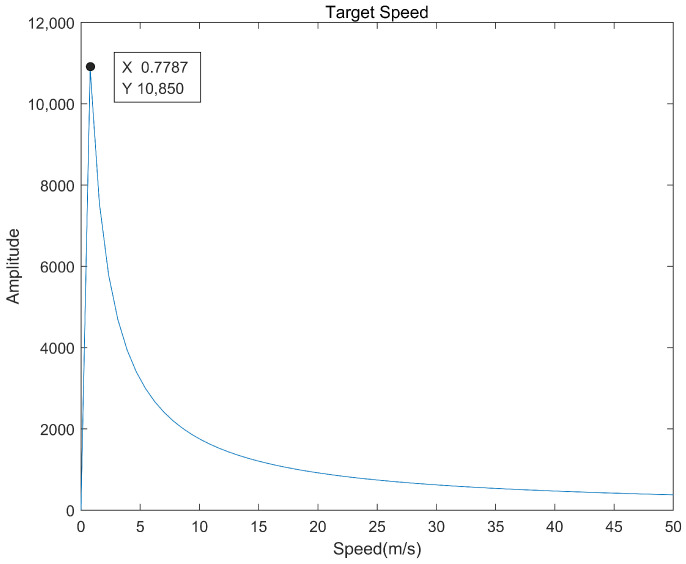
Simulation results of moving targets.

**Figure 4 sensors-22-04535-f004:**
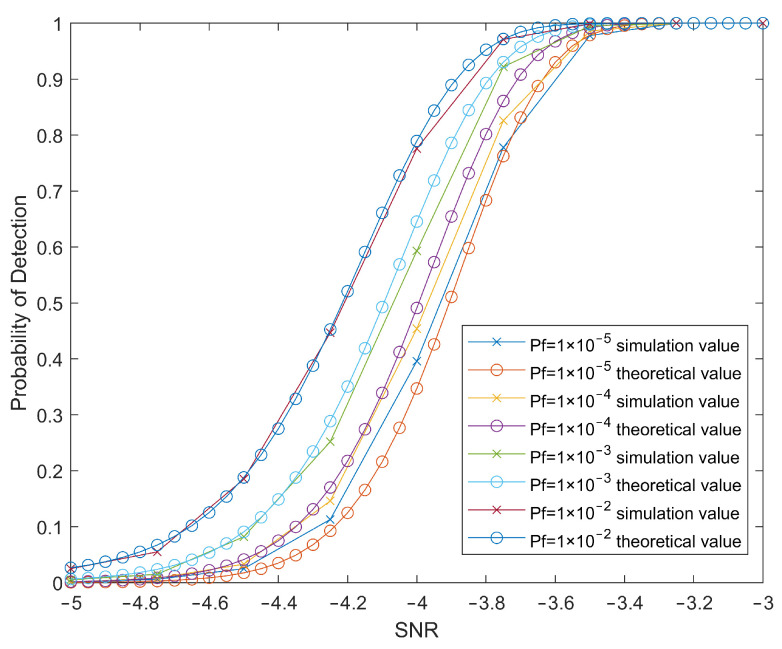
Target detection probability under different $P_{f}$.

**Figure 5 sensors-22-04535-f005:**
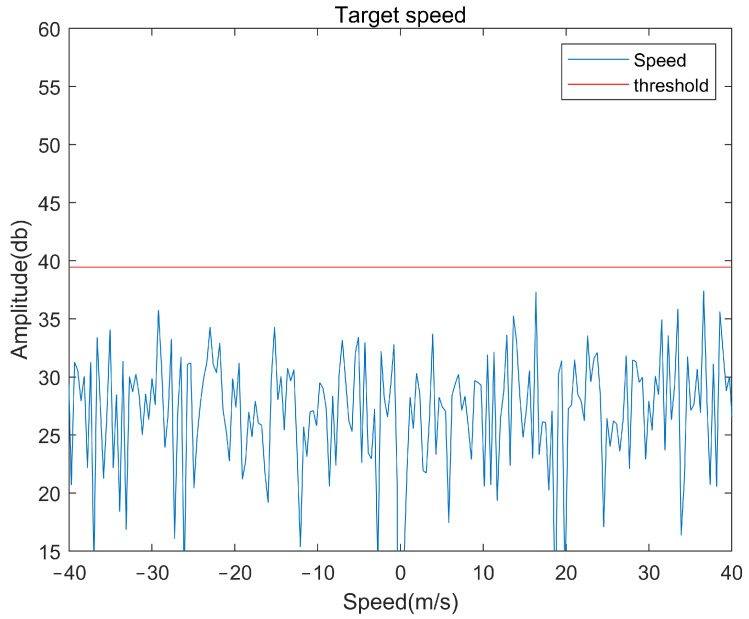
Target detection results in the presence of clutter interference.

**Figure 6 sensors-22-04535-f006:**
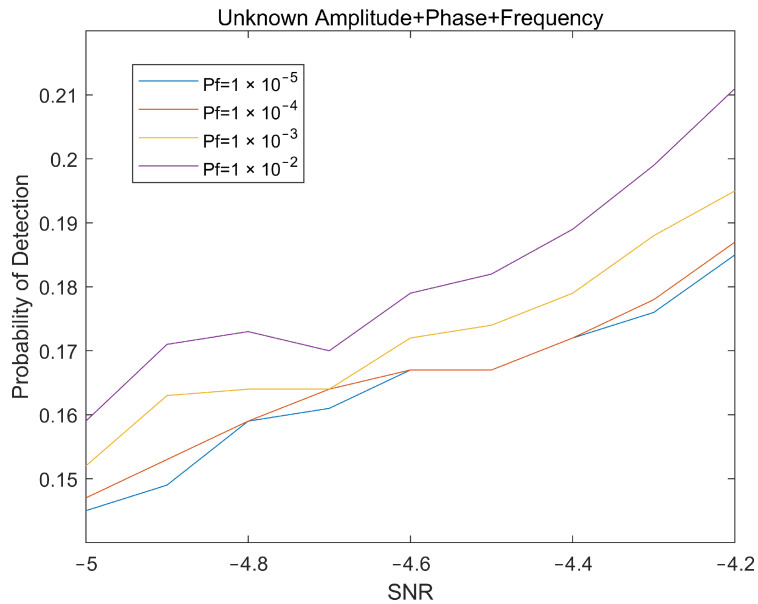
Target detection probability under different $P_{f}$ in the presence of clutter interference.

**Figure 7 sensors-22-04535-f007:**
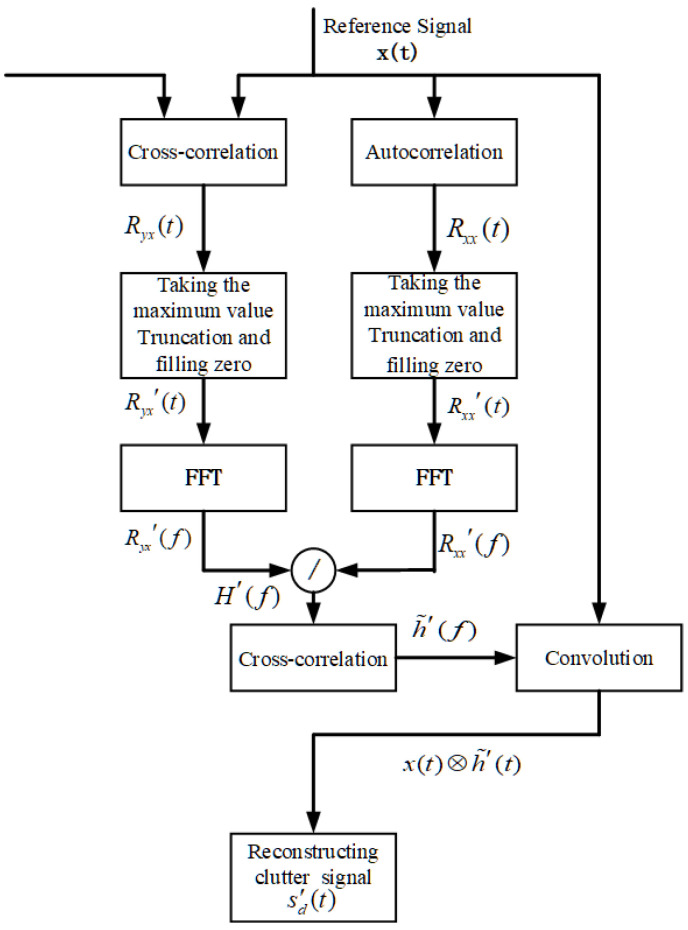
Processing of the deconvolution algorithm.

**Figure 8 sensors-22-04535-f008:**
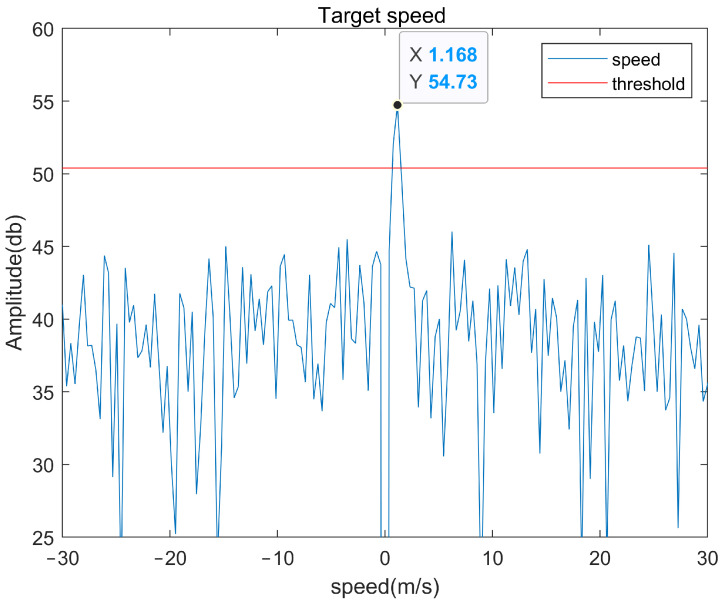
Target detection results after clutter processing.

**Figure 9 sensors-22-04535-f009:**
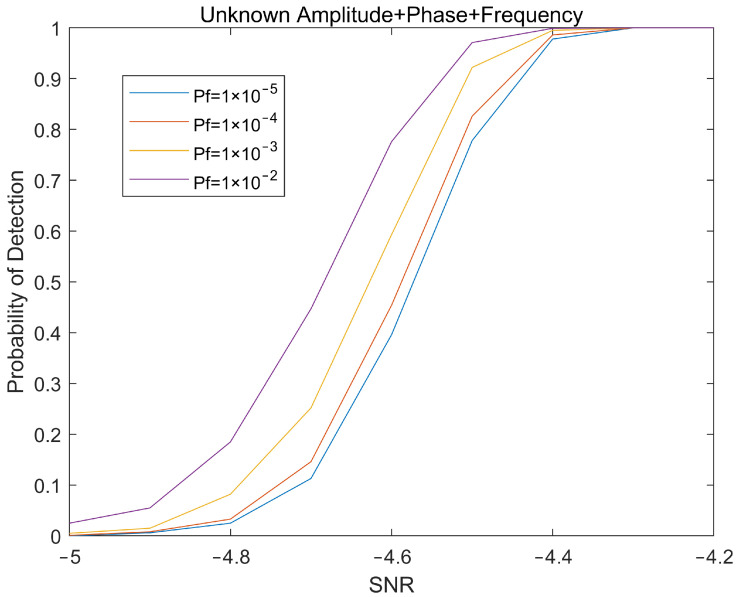
Target detection probability under different $P_{f}$ after clutter processing.

**Figure 10 sensors-22-04535-f010:**
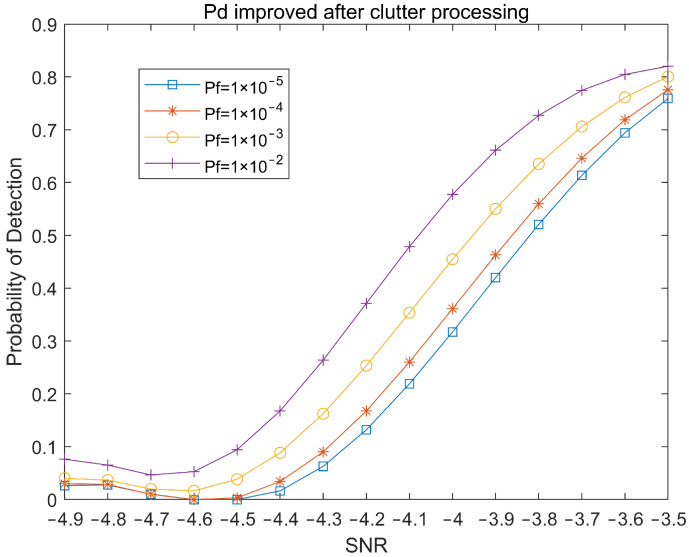
Improved detection probability of the system after clutter processing.

**Figure 11 sensors-22-04535-f011:**
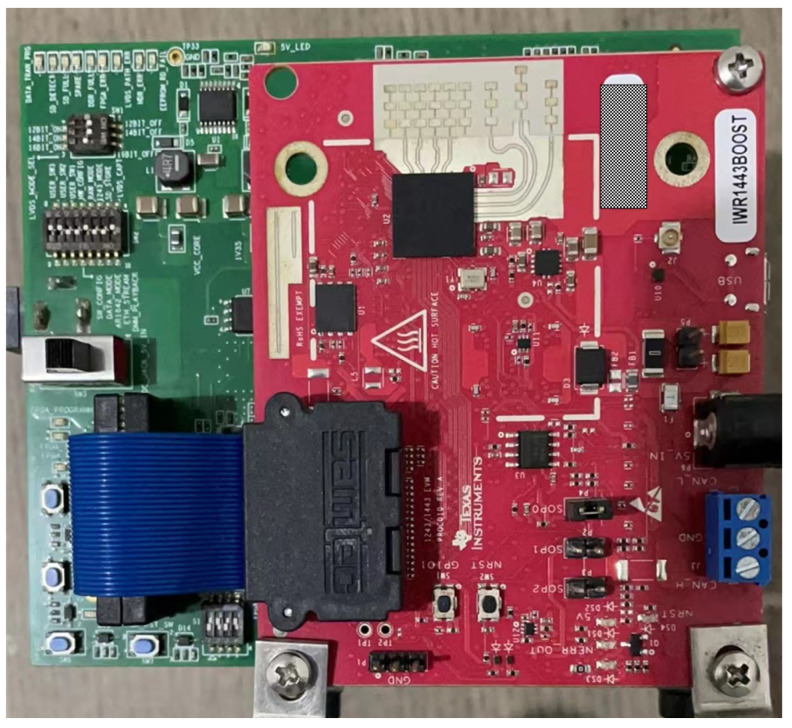
Radar equipment used in the experiment.

**Figure 12 sensors-22-04535-f012:**
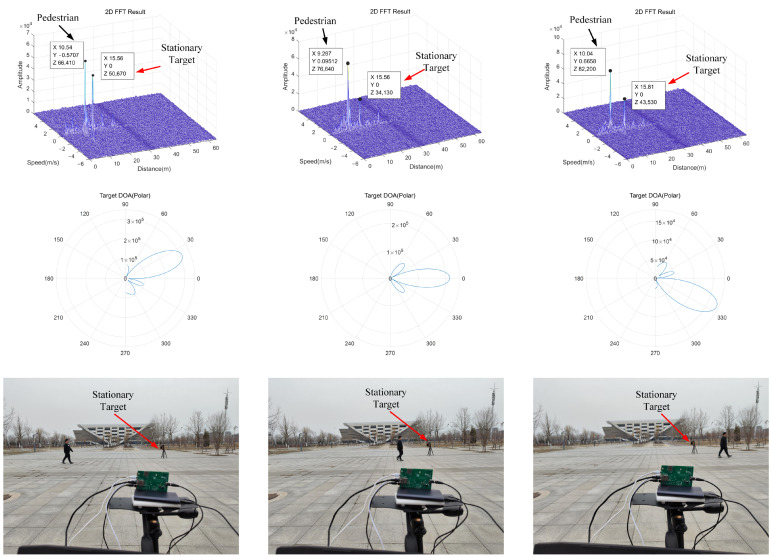
Pedestrians walk in the direction perpendicular to the radar LOS.

**Figure 13 sensors-22-04535-f013:**
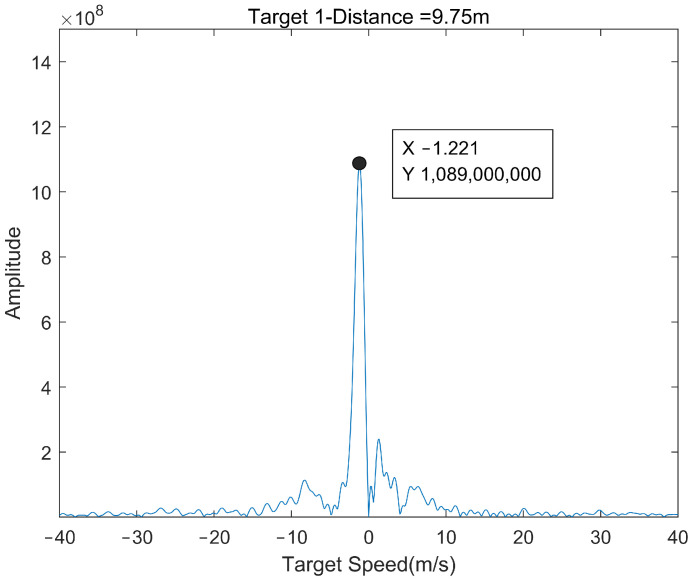
The final measured speed of pedestrians.

**Figure 14 sensors-22-04535-f014:**
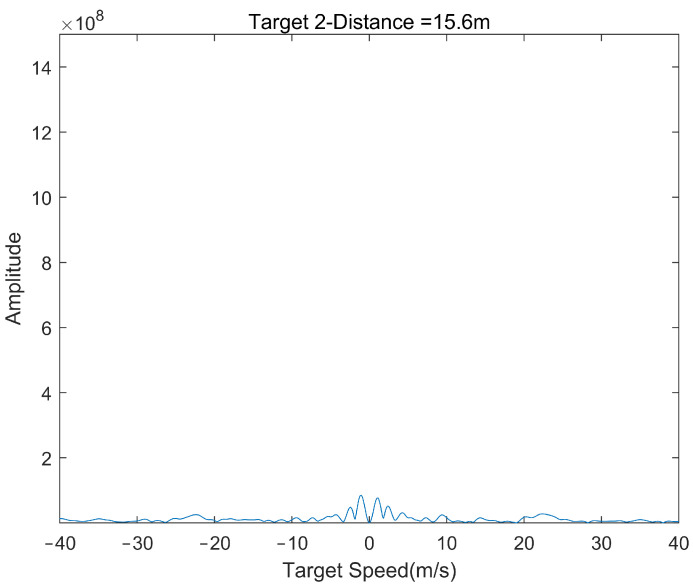
The final measured velocity of the stationary target.

**Table 1 sensors-22-04535-t001:** Simulation experimental radar parameters.

Parameter	Value
frequency (GHz)	77
Pulse width (s)	1/1,000,000
Pulse number	1,000,000
Bandwidth (MHz)	150
Range resolution (m)	1
Velocity resolution(m/s)	0.2
Target speed (m/s)	1

**Table 2 sensors-22-04535-t002:** Radar parameters actually tested.

Parameter	Value
Carrier frequency (GHz)	77
Bandwidth (MHz)	150
Sampling frequency	10${}^{6}$
Number of transmitted frames	200
Number of cycles per frame	128
Frame transmission period (s)	0.07
Range resolution (m)	1
Velocity resolution (m/s)	0.5

## Data Availability

The datasets generated from this study are available upon reasonable request.

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
