# Peer review of "A New Side-Looking Scheme for Speed Estimation and Detection of Tangential Slow-Moving Targets"

_sensors, 2022, doi:10.3390/s22124535_

Round 1

Reviewer 1 Report

This paper proposed a multi-channel radar scheme and a new signal processing method to realize the target speed estimation and detection with tangential velocity or weak radial velocity. The main contributions of this paper can be summarized as follows:

(1)     A side-looking radar scheme is proposed, which uses the change frequency of the wave-path difference between the receiving channels to replace the Doppler frequency to estimate the velocity of the tangential moving targets.

(2)     A deconvolution-based clutter map algorithm is proposed to solve the problem of clutter interference in the test environment;

There are some questions need to be clarified before the paper publication:

(1)       Clutter map CFAR is time domain clutter suppression method, so please check the statement of line 208.

(2)       Please check equation(20) to make sure it is correct.

(3)       What is the physical meaning of the clutter introduced by the transmission channel? Usually we only consider the channel noise.

(4)       Please give a more detailed description of the clutter suppression chapter. I feel confused with this chapter.

Reviewer 2 Report

  1. Radar is the primary detection sensor used in both military and civilian applications. This article presents a method for improving the speed estimation and detection by radar system.
  2. General remarks
    1. Too many abbreviations make it difficult to follow the content of the article. Each abbreviation should be expanded the first time it appears. Not all readers need to know all abbreviations. Especially in the abstract of the article and conclusions.
    2. Please use the language of a scientific research report without personal references like “we” or “our”.
    3. The final conclusions are too general and only generally summarize the research presented in the article. I suggest expanding the conclusions with more detailed findings.
    4. The subject of object tracking is commonly associated with radar tracking, also Frequency Modulated Continuous Wave (FMCW) radars. It is worth to compare results presented in the article with papers listed below. Publications worth analyzing include:
      • Stateczny, A.: Radar Water Level Sensors for Full Implementation of the River Information Services of Border and Lower Section of the Oder in Poland. Proceedings of 17th International Radar Symposium (IRS), International Radar Symposium Proceedings, Krakow, Poland (2016), 10.1109/IRS.2016.7497386.
      • Nowak, A.; Naus, K.; Maksimiuk, D., A method of fast and simultaneous calibration of many mobile FMCW radars operating in a network anti-drone system. Remote Sensing, vol.11, issue 22, art. 2617, 2019, doi:10.3390/rs11222617
      • Lubczonek, J,: Analysis of accuracy of surveillance radar image overlay by using georeferencing method. International Radar Symposium Proceedings, pp. 876 – 88126, article number 7226230, Dresden, Germany (2015), doi: 10.1109/IRS.2015.7226230.
      • Stateczny, A., Lubczonek, J.: FMCW Radar Implementation in River Information Services in Poland. International Radar Symposium Proceedings, pp. 852-857, Dresden, Germany (2015), doi: 10.1109/IRS.2015.7226245.

5. However the article is very well written should be carefully edited. Some remarks included below.

  1. Specific remarks
    1. The article should be edited more carefully, in lines 156-165 we have different font and no justification.
    2. The conditions of the experiment should be described in more detail.
    3. What means speed about -1. 2m/s, line 321. Speed cannot be negative.
    4. What means Y 1089000000 on the Fig.12?

Reviewer 3 Report

Review comments on the manuscript by Qi et al 

A new side-looking scheme for speed estimation and detection of tangential slow-moving targets

The aim of the study was to describe the methodology for a new side-looking scheme for speed estimation and detection of tangential slow-moving targets. The authors propose and describe the details of a side-looking radar velocity measurement system that utilized a new signal processing method and a multi-channel radar scheme. They also describe a deconvolution-based clutter map algorithm to reduce the issue of background clutter and test and verify its performance using simulations. The paper describes the methodological changes that the author made to the existing procedures and describes how each modification improves the outcomes of the analysis. A 77G commercial radar was used to test the proposed new system and demonstrated that the clutter algorithm can more effectively detect and accurately estimate the speed of tangential low-speed targets under clutter interference compared with previous methods.

Overall, the data presented in the paper achieves the study aims and provides an original contribution to improving our knowledge and ability on procedures for use of radar systems to measure the speed of moving targets. The statistical basis for the changes to image analysis techniques were appropriate to address the aims of the study. As such, I consider that a revised paper incorporating the marked comments on the attached PDF and addressing the comments detailed below could be resubmitted for assessment for publication.

Given that the author’s first language is not English, I have marked on the PDF document, numerous suggested spelling and grammatical changes to improve the quality and flow of the manuscript. The authors should review these changes and have the manuscript reviewed before resubmission.

Specific comments on the manuscript are detailed below.

There are several issues with the references. For some references, the first letter of each word in the paper title is capitalised. The authors should check previous issues of the journal regarding presentation of references.

Round 2

Reviewer 2 Report

The authors have revised the paper in accordance with the reviewers instructions and the paper can now be published.